# "The worst thing is lying in bed thinking 'I want a cigarette'" a qualitative exploration of smoker's and ex-smoker's perceptions of sleep during a quit attempt and the use of cognitive behavioural therapy for insomnia to aid cessation

Joe A. Matthews[1,2,3,4]*, Victoria R. Carlisle[5ʘ], Robert Walker[5ʘ], Emma J. Dennie[1,3,6ʘ], Claire Durant[1‡], Ryan McConville[7‡], Hanna K. Isotalus[2‡], Angela S. Attwood[1,3,4ʘ]

1 School of Psychological Science, University of Bristol, Bristol, United Kingdom, 2 Department of Electrical and Electronic Engineering, University of Bristol, Bristol, United Kingdom, 3 Medical Research Council Integrative Epidemiology Unit, University of Bristol, Bristol, United Kingdom, 4 Integrative Cancer Epidemiology Programme, University of Bristol, Bristol, United Kingdom, 5 Department of Population Health Sciences, Bristol Medical School, University of Bristol, Bristol, United Kingdom, 6 University of Bristol Business School, University of Bristol, Bristol, United Kingdom, 7 Department Engineering and Mathematics, University of Bristol, Bristol, United Kingdom

ʘ These authors contributed equally to this work.
‡ These authors also contributed equally to this work
* Joe.matthews@bristol.ac.uk

## Abstract

Smokers report poorer sleep quality than non-smokers and sleep quality deteriorates further during cessation, increasing risk of smoking relapse. Despite the use of cognitive behavioural therapy for insomnia (CBT-I) to aid quit attempts emerging in the area, little is known about smokers and ex smoker's experiences of sleep during a quit attempt or their perceptions of CBT-I. This study addresses this gap by exploring smoker's and ex-smoker's experiences of the link between smoking and sleep and how this may change as a function of smoking/smoking abstinence. It also explores views of traditional CBT-I components (i.e., perceived feasibility, effectiveness, barriers of use). We conducted semi-structured interviews with current and recently quit smokers (n = 17) between January and September 2022. The framework method was used for analysis. Four themes addressing research questions were described. These included: 1) A viscous cycle; poor sleep quality and negative psychological state during cessation; 2) Perceived engagement and effectiveness; the importance of feasibility, experience, value, identity and psychological state in assessing CBT-I as a cessation tool; 3) Striking a balance; tailoring CBT-I to reduce psychological overload in a time of lifestyle transition; and 4) Personalisation and digital delivery helping overcome psychological barriers during cessation. The analysis suggested during quit attempts smokers experienced a range of sleep problems that could increase risk of relapse due to a negative impact on psychological state. It also revealed participants thought that CBT-I is something they would use during a quit attempt but suggested changes and

**Data Availability Statement:** Data are available at the University of Bristol data repository, data.bris, at https://doi.org/10.5523/bris. 1mfv9zsgcxojo2otfcw0d23ifd.

**Funding:** This research was funded by Cancer Research UK (CRUK) (C18281/A29019) https:// www.cancerresearchuk.org/funding-for-researchers (AA) and the Engineering and Physical Sciences Research Council (EPSRC) (EP/S023704/ 1) https://www.ukri.org/councils/epsrc/ (JM). The funders had no role in study design, data collection and analysis, decision to publish, or preparation of the manuscript.

**Competing interests:** The authors have declared that no competing interests exist.

additions that would improve engagement and be better tailored to quitting smokers. Key additions included the integration of smoking-based cognitive restructuring, starting the intervention prior to a quit attempt, and the need for personalisation and tailoring.

## Introduction

Smoking continues to be the leading cause of death and morbidity worldwide [1]. Despite the negative health consequences over 1.3 billion individuals smoke globally [2]. Only one in twenty smokers who engage in a quit attempt will remain smoke free [3] and 80% of quit attempts will end in relapse within six months [4]. Therefore, there is a need to explore new strategies and mechanisms to aid individuals quitting smoking.

Poor sleep quality, both before and during cessation, increases the risk of relapse during a quit attempt [5]. There are a few plausible mechanisms in which poor sleep quality impacts a cessation attempt. Previous work has found poor sleep quality is associated with an increase in urge to smoke [6], lower abstinence self-efficacy [7] and decision making [8]. A reason for this may be that poor sleep quality or duration leads to emotion dysregulation, which may in turn, lead to reductions in self-efficacy and other relapse risk factors [9]. Another hypothesis is that fatigued smokers may look to the stimulant effect of nicotine as a coping strategy [5]. To the authors' knowledge no studies have explored smokers' perceptions of sleep during cessation and explored these as a potential mechanism of relapse. Given the strong relationship between sleep quality and risk of relapse, interventions to improve sleep either prior to or during a quit attempt offer a promising avenue of research.

While sleep may be a potential modifiable target to aid smoking cessation, at the time of writing, few studies have targeted sleep in quitting smokers. Those that have employed Cognitive Behavioural Therapy for Insomnia (CBT-I), or components of this program, to improve sleep outcomes in quitting smokers; have reported mixed results [10,11]. CBT-I is a treatment for chronic insomnia that has been found to be more effective than pharmacological support [12,13] and has been used successfully in a variety of populations including chronic pain [14], psychiatric disorders [15,16] and cancer patients [17]. It typically comprises of four to five components including sleep-hygiene/ stimulus control, sleep restriction therapy, relaxation training and cognitive restructuring. CBT-I has shown effectiveness using multiple delivery methods including face-to-face, in a group setting [18] and more recently interactive digital platforms [19].

Sleep hygiene and stimulus control address physiological, environmental and behavioural factors (e.g., limiting alcohol and caffeine consumption, introducing regular sleep schedule and no screens in the bedroom), as well as attempting to strengthen the association between the bed, the bedroom and sleep. Sleep restriction is the manipulation of homeostatic sleep drive by initially reducing time in bed to limit an individual's sleep opportunity to their average sleep duration. Time in bed is then adjusted weekly based on the individual's sleep efficiency of the previous week, with the aim of gradually increasing sleep duration. Relaxation training can take many forms, including but not limited to; deep breathing, progressive muscle relaxation and meditation. It aims to negate cognitive, emotional or psychological arousal, both in the day and at night. Finally, cognitive restructuring, aims to identify and correct dysfunctional thinking patterns an individual may have about sleep, insomnia and fatigue.

Despite evidence supporting a link between sleep and smoking, and the use of CBT-I as an intervention to aid cessation, there has been little research on smoker's perceptions of sleep quality in relation to a quit or CBT-I as an intervention to support quit attempts. Without

understanding smoker's experiences of sleep and potential barriers to engaging with interventions such as CBT-I, the effectiveness of such programs may be compromised by failure to engage with user-centred development frameworks [20]. Many intervention development frameworks have qualitative research at their core [21] which can improve both acceptability and feasibility of interventions, by understanding the experiences, needs and goals of the target user. This study recruited smokers and smokers that had recently quit to explore their perspectives on sleep during cessation and the feasibility of traditional CBT-I components within the context of a quit attempt.

The primary aim of this study was to explore how a CBT-I intervention may be adapted for quitting smokers. To do this we explored smoker's and ex-smoker's views (i.e., perceived feasibility, effectiveness, barriers of use) of traditional CBT-I components. To better understand the context in which an intervention would be delivered, we also explored smoker's and ex-smoker's perceptions and experiences of the link between smoking and sleep and how this may have changed as a function of smoking/smoking abstinence.

To achieve this, we conducted semi-structured interviews with current smokers that had previously engaged in a quit attempt and recently quit (<12 months/currently quitting). The research aims that guided the semi-structured interviews are outlined below; however, the interview guide was used to explore specific areas in more depth.

1. Explore potential barriers and facilitators to engagement in traditional CBT-I components to facilitate intervention development.

2. Identify potential changes and/or additions smokers and ex-smokers would like to see to existing CBT-I frameworks.

3. Understand smokers/ ex-smokers perceived impact of a cigarette abstinence on sleep quality and impact of poor sleep quality on a quit attempt.

## Methods

### Design

We conducted semi-structured interviews with current smokers and ex-smokers. This study was underpinned by critical realism [22]. Critical realism is a meta-theory that posits that reality exists independent of human beings, but our observations of this reality are mediated by language, culture, and history and are subsequently relative in nature. Ethics approval was obtained from the School of Psychological Science Research Ethics Committee at the University of Bristol (REF: 9130). The study protocol was pre-registered on the open science framework (osf.io/cr4qv/).

### Participants, recruitment, and sampling

Convenience sampling was employed to recruit participants by advertising both online (e.g., smoking cessation groups) and within the community, including health centres and pubs. Study adverts were also posted online in a variety of smoking-based forums. Adverts directed individuals to contact the study team via email if they were interested in taking part.

Inclusion criteria were: 18 years or older, regular smoker (≥5 cigarettes per day for at least 3 months) who has previously engaged in a quit attempt or recently quit/quitting smoker (quit within last 12 months) and had English as a first language or similar level of fluency. Our recruitment target was 20 participants (in line with data saturation recommendations by Green and Thorogood (2004)), but this was flexible based on the reflective nature of "information power" items such as specificity of study aims, strength of dialogue and theory [23]. In total we recruited 17 participants (15 current smokers and 2 ex-smokers).

## Interviews

Semi-structured interviews were conducted between January and September 2022 using an interview guide that was developed by the study team, reviewed by current smokers (n = 2) and refined throughout the process. The interview guide (S1 File) was split into two sections. Section 1 began with a broad question on participants' sleep history, sleep during previous quit attempts and consequences of poor sleep during a quit attempt. In section 2, the researcher read from a script providing participants with brief information of the relationship between sleep and smoking and then played a 12-minute video showing fictional patient Theo's journey through each component of CBT-I. After the video, participants were asked questions about their perceptions of CBT-I components: feasibility during a quit attempt, barriers and facilitators to engagement, any suggestions or modifications applicable to a quit attempt or smoker and potential unintended consequences of engagement. Finally, participant's views on CBT-I being delivered digitally were also explored. To allow participants to shape the direction of the interview, questions were phrased in an open and non-leading manner, and interesting responses were followed up with additional questions.

All interviews were conducted online (https://zoom.us/) by the first author (no other non-participant attendees). Audio recordings were downloaded (Duration M = 43 minutes 42 seconds) and transcribed verbatim by the first author (For research team an reflexivity information see S1 File). Participants received a £10 gift voucher following the interview as recognition of their time.

## Demographics

The demographic survey contained questions on age, sex, current smoking status, ethnicity, highest qualification attained, smoking history, quit-attempt history, e-cigarette use, cessation medication during quit attempts and cigarettes smoked per day. Additionally participants completed the Fagerström Test of Nicotine Dependence (FTND) [24], Pittsburgh Sleep Quality Index (PSQI) [25] and readiness to quit ladder[26] (see description below).

Sleep Quality: The Pittsburgh Sleep Quality Index [25] has been widely used in the context of sleep and behavioural research. Nineteen items generate a global sleep quality score. Validity and reliability of this instrument are reported elsewhere [25,27]. Global sleep quality scores range from 0 to 21, with higher scores indicating worse sleep quality.

Readiness to Quit: Participants completed the Readiness to Quit Ladder [26], which is single item measure of motivation to change smoking behaviour. It uses a 10-point ordinal scale with responses ranging from 1 = "I have decided to continue smoking" to 10 = "I have already quit smoking." This instrument performs well when predicting smoking rate, quit attempts and cessation, and is associated with cognitive and behavioural indicators of readiness to consider smoking abstinence [26,28].

Nicotine Dependence: Fagerström Test of Nicotine Dependence [24] is a validated questionnaire containing six items that are closely related to biochemical indices of nicotine dependence. Scores range from 0 to 10 points, with higher scores indicating higher nicotine dependence. Dependence categories are as follows; less than 4 points = minimally dependent, 4–6 points = moderately dependent, 7–10 points = highly dependent.

To contextualise the data, participants' nicotine dependence and sleep quality score are provided alongside each quote.

## Data analysis

Prior to anonymisation and analysis, all transcripts were checked for accuracy by a second member of the research team (ED). Anonymised transcripts were entered into QSR Nvivo 12 [29] for analysis. Interview data were thematically analysed using the framework method [30].

The framework method was selected as it allows multidisciplinary teams with diverse research experience and expertise to engage with qualitative data using the summarise matrix [30]. This approach incorporates multiple stages of analysis: familiarisation; coding; framework development; framework application; and interpretation, allowing themes to be generated both inductively from personal views and experiences and deductively from theoretical constructs or existing literature.

For familiarisation, two researchers (JM, ED) read the transcripts to immerse themselves in the data. A draft framework was developed by both researchers after independently reviewing a sub-set of the transcripts (n = 4) and generating codes in relation to our research questions. Researchers met after coding a further two different transcripts to discuss coding and refine the framework if necessary. This process continued until no other changes to the framework were required and the final framework could be applied to all the transcripts. The researchers met regularly to ensure the framework was being applied consistently and to discuss coding. If any disagreements about coding occurred other members of the research team were consulted (VC and RW). After coding researchers met (JM, ED, VC) to discuss themes and sub themes. Summaries for each of these were detailed, and quotes extracted to aid write up.

## Results

Table 1 outlines a subset of participant demographic and characteristic information (Full demographic information can be found in (S1 Table). Participants (n = 17) were mostly female (n = 9) and aged between 22 and 62 years (M = 38.8, SD = 13.32). Current smokers (n = 15) reported smoking between 6 and 30 cigarettes per day (M = 13.8, SD = 7.4) and showed low to high levels of nicotine dependence, scoring between 3 and 8 (M = 4.3, SD = 1.7) on the FTND. On the PSQI participants scored between 4 and 16 (M = 7.5, SD = 3.4) and 88% of participants scored $\geq$ 5 which is categorised as poor sleep. None of the study participants were aware of or had experience of CBT-I prior to the interview.

**Table 1. Participant demographic and characteristics.**

| Participant | Sex | Age | Smoking Status | Nicotine Dependence | Previous quit attempts | PSQI |
|---|---|---|---|---|---|---|
| 001 | Female | 31 | Smoker | Low to moderate | One | 7 |
| 002 | Male | 25 | Smoker | Low to moderate | Two | 16 |
| 003 | Male | 25 | Smoker | Moderate | Two | 14 |
| 004 | Female | 43 | Smoker | Moderate | Four | 8 |
| 005 | Female | 28 | Smoker | Low to moderate | Three | 8 |
| 006 | Female | 25 | Non-smoker | N/A | Five or more | 8 |
| 007 | Male | 60 | Smoker | Moderate | Five or more | 5 |
| 008 | Female | 43 | Smoker | Low to moderate | Five or more | 5 |
| 009 | Male | 28 | Smoker | Low | Two | 4 |
| 010 | Female | 28 | Smoker | Low | Five or more | 6 |
| 011 | Male | 29 | Non-smoker | N/A | Five or more | 5 |
| 012 | Female | 46 | Smoker | High | Three | 12 |
| 013 | Female | 26 | Smoker | Low to moderate | Three | 11 |
| 014 | Male | 31 | Smoker | Moderate | Two | 4 |
| 015 | Male | 22 | Smoker | Low to moderate | Two | 8 |
| 016 | Male | 57 | Smoker | High | Four | 7 |
| 017 | Female | 62 | Smoker | Moderate | Three | 5 |

We developed four themes in relation to our participants experiences and perceptions of sleep during smoking cessation and CBT-I as a method of aiding a quit attempt: An overview of these themes shown in Table 2.

## A vicious cycle; poor sleep quality and psychological state during cessation

This theme depicts a cycle during cessation that revolved around poor sleep and mental health. A sense that a quit attempt led to a sudden decline in psychological state and poor sleep was evoked in the data. Nicotine withdrawal was described as an immediate "shock" to the body, which increased feelings of worry and stress and ruminating thoughts about smoking. For many, this led to an abrupt reduction in sleep quality during a quit attempt that was characterised by reduced sleep duration, increased time taken to fall asleep, and increased sleep disruption.

*"At first it was pretty terrible actually [sleep] and I think that was, I smoked for 16 years and that initial eradication of any nicotine was quite a shock to my body" (PID 011, Ex-smoker, PSQI 5).*

*"Initially, it took me longer to go to sleep because I'd be thinking about cigarettes. That's on the serious occasions I've tried to stop" (PID 007, Moderately dependent smoker, PSQI 5).*

*"I woke up frequently in the sleep and then when I wake up, going back to sleep now becomes a problem" (PID 003, Moderately dependent smoker, PSQI 14).*

The relationship between psychological state and sleep quality did not appear to be unidirectional, but cyclical; as poorer sleep quality led to poorer mood, unease and stress which led to a reduction in sleep quality. This created increased feelings of anxiety surrounding sleep for some participants, as they felt quality sleep played a pivotal role to improving their mental resilience, yet knew this would be challenging during a quit attempt.

**Table 2. Research questions addressed, theme names and definitions.**

| Research question | Themes that address research question | Theme description |
|---|---|---|
| **RQ 3** | A viscous cycle; poor sleep quality and psychological state during cessation | This theme explores experience of sleep during smoking cessation among current smokers and ex-smokers. Highlighting, a cycle of poor sleep impacting psychological state and vice versa. Revealing a conscious value prioritisation between both behaviours. |
| **RQ 1 + 2** | Perceived engagement and effectiveness; the importance of feasibility, experience, value, identity and psychological state in assessing CBT-I as a tool for cessation | This theme identifies foundations of perceived likelihood of engagement with CBT-I, highlighting the importance of previous experiences, identity, perceived value and psychological state associated with cessation. |
| | Striking a balance; tailoring CBT-I to reduce psychological overload in a time of lifestyle transition | This theme depicts the sense of overload when undertaking multiple changes during cessation and uncovered ways to adapt traditional CBT-I structure and components to reduce impacts on self-efficacy and increase the likelihood of adherence and efficacy. |
| | Personalisation and digital delivery helping overcome psychological barriers during cessation | This theme investigates how digital delivery and personalisation may reduce the mental burden of patients in an already deteriorated psychological state and how traditional barriers to intervention engagement may be removed. |

*"it's a vicious cycle isn't it? Because you get in your own head and then you're thinking oh I've got to be up in five hours, then you get stressed about that. Then you don't sleep even more, and it was just making me quite anxious about sleep"*

*(PID 011, Ex-smoker, PSQI 5).*

This cyclical relationship created an extremely challenging period. Some participants recalled feeling like they were fatigued and cognitively underperforming, increasing the likelihood of making a bad decision or contributing to the desire to smoke to alleviate such side effects. Factors such as fatigue, decision making and reduced cognition that are associated with poor sleep, were highlighted as issues that were perceived to be alleviated by a return to smoking, further increasing risk of relapse.

*"If I'm more tired, I'm more likely to make what I would later reflect on as a bad decision or something I wouldn't have done" (PID 006, Ex-smoker, PSQI 8).*

An interesting paradigm of value prioritisation between the behaviours of sleep, smoking, and mental health emerged. Participants highlighted the impact both behaviours played in decision making in relation to one another. Some participants regularly and consciously reduced the duration of their sleep opportunity to prioritise smoking, while others did not feel quitting smoking was worth the subsequent reduction in sleep quality and psychological state that could impact many other facets of their lives. Potentially suggesting a prioritisation of short-term mental wellbeing over longer-term health impacts.

*"I try to go to bed as late as possible. So I can have that last fag. . .a little bit later"*

*(PID 001, Low to moderately dependent smoker, PSQI 7).*

*"If all of a sudden you have to go three months without sleep. For me that would be a big reason not to give up. I'd rather smoke and get a good night's kip. Then have to fight through three months of being a little bit of a zombie. Not quite on your game" (PID 004, Moderately dependent smoker, PSQI 8).*

However, sleep quality and its perceived links with better mood, improved cognition and reduction in stress led to many participants concluding that if they had slept better during their quit attempt(s), they may have been more resilient to the side effects and challenges associated with smoking cessation and less likely to relapse.

*"I didn't really have a good night's sleep for a while. [. . .] Just my mental state as well went massively downhill. [. . .] If I had, had a better night's sleep. I'd probably would have coped a bit better. But yeah, my head was just gone. So four months I was like 'right need a fag'" (PID 001, Low to moderately dependent smoker, PSQI 7)*

### Perceived engagement and effectiveness; the importance of feasibility, experience, value, identity and psychological state in assessing CBT-I as a tool for cessation

This theme reflects participants' perspectives on how CBT-I could be used as a tool for cessation. Multiple factors impacted perceived effectiveness of, and engagement with, CBT-I components, such as feasibility of adherence, value, identity and previous experiences of quit attempts.

Quitting smoking is a challenging period for many. The belief that components of CBT-I were achievable was important for perceived engagement, and contributed to participant's general view of CBT-I as a potentially effective aid to smoking cessation.

*"There was lots of things I can't remember them all but there were lots of things, when they were coming up. Those changes, I was like. . . yes, yes, yes. Things that I believe I could do"*

*(PID 008, Low to moderately dependent smoker, PSQI 5).*

Perceived value was also a key determinant of how participants viewed CBT-I as a whole or its individual components. For example, the value and utility of cognitive restructuring was identified by nearly all participants. Some participants identified that this component may help attenuate the "rumination" associated with getting to sleep, which they have experienced in previous quit attempts.

*"I think that would help if I was struggling. I think restructuring undesired thinking patterns, I mean when I'm sleeping well, I don't really have those thoughts, but when I can't sleep, like everyone does, I start having those thoughts"*

*(PID 014, Low to moderately dependent smoker, PSQI 4).*

In summary, the value and feasibility of a component were identified as important, and lack thereof could act as barrier to engagement. Challenging thought patterns around sleep could be particularly difficult during a quit attempt, as trying to restructure thoughts at a time of high stress and poor mood would be difficult.

Distraction was identified as an important strategy during a quit attempt, and several participants judged CBT-I components on the basis of whether they positively or negatively impacted distraction strategies. For example, relaxation training was perceived as a positive distraction technique, whereas both sleep restriction and removal of the TV from the bedroom (stimulus control) were identified as major inhibitors of distraction. Sleep restriction was perceived to potentially encourage smoking based thoughts by increasing craving and poor mood as patients would have more waking hours than usual. Restricting of sleep or a ban of devices during times of high likelihood of rumination and craving could have led to increased risk of relapse.

*"I think the difference I suppose with the quit effect is that sometimes you want to go to bed earlier because you want the day to be over. You can't smoke when you're sleeping. You just want to get to bed and then you can toss and turn and toss and turn. So that might be challenging"* (PID 004, Moderately dependent smoker, PSQI 8).

*"So, the thing with me and smoking is, it's yeah, I do get the cravings, but a lot of the time it's also boredom as well and keeping myself pre-occupied and keeping my hands busy. Doing whatever, so I'm not, 'I'm sitting here I've got nothing to do, I'll just roll a cigarette [. . .] Like not watching TV in my room, I'd just think about smoking if I wasn't doing that. Plus, my Dad smokes, so if I had to spend more time down there, that would make me want a smoke. Whereas upstairs I'm away from that trigger"* (PID 015, Low to moderately dependent smoker, PSQI 8).

Personal preferences and identity played an important role in engagement, often more so than the perceived value, feasibility or effectiveness of a component. For example, many participants believed relaxation training to be a feasible and effective method both to aid sleep and

potentially cessation. However, some of these individuals reported it would not be effective for themselves, as they described relaxation training in terms such as "hippy" and "airy fairy", which they did not personally identify with. Therefore, overcoming some of these preconceived stereotypes may be important in engaging certain groups.

*"Relaxation not for everybody but I imagine would work for a lot of people, me myself probably not, I did try it and it did work for a little bit but you know when you feel a bit silly doing it?" (PID 015, Low to moderately dependent smoker, PSQI 8).*

*"I just don't see myself doing these relaxation things. It's almost a bit like doing a meditation, it makes me feel a bit hippy" (PID 017, Moderately dependent smoker, PSQI 5).*

### Striking a balance; tailoring CBT-I to reduce psychological overload in a time of lifestyle transition

This theme depicts the participants sense of apprehension at taking on so many changes in their lifestyle within the context of significant emotional challenges associated with smoking cessation, while also hypothesising that such a big change in lifestyle may increase the likelihood of cessation. Timing of CBT-I delivery was seen as one of the most important indicators of adherence and potential success in conjunction with a quit attempt and ultimately of the quit attempt itself. Starting CBT-I and a quit attempt at the same time was described as a recipe for failure and unfeasible by nearly all participants. Nearly all suggested CBT-I should be started a month prior to a quit attempt.

*"I think if you try and do both at the same time. You will have a very small chance of success" (PID 011, Ex-smoker, PSQI 5).*

*"It would be difficult to change all at once. Because I get, as I said before when you stop smoking you feel as though you are depriving yourself of something. But then to try and change all of them at the same time as well. You'll feel as though you're depriving yourself of something else." (PID 007, Moderately dependent smoker, PSQI 5).*

The importance of routine in every facet of behaviour and behaviour change was a common narrative. Participants explained that changing to a routine and adhering to it prior to quitting, slowly alters your mindset so you feel more capable, positive and have greater self-efficacy to undertake a quit attempt.

*"Because what it is, is it's a routine and it's building good habits. It's a good routine it's a positive routine and it's just all part and parcel of breaking that habitual thing surrounding smoking [. . .] I think in that four-to-six-week journey, you could put yourself in that positive state of mind to feel that you could actually do something about smoking. Because there is no such thing as quitting without will power. You know it just doesn't exist" (PID 017, Moderately dependent smoker, PSQI 5).*

However, while a new routine was perceived as a gateway to a successful quit attempt, choice in CBT-I component to reduce the likelihood of failure was highlighted again by many participants. Making CBT-I achievable on an individual basis appears to be necessary to facilitate a new routine and aid the quit attempt. This is a difficult challenge for intervention design. Removal of too many components could lead to a reduced effectiveness but our data suggests lack of intervention flexibility could impact self-efficacy in the weeks prior or during a quit attempt.

*"I think if I did all of them, even the ones I didn't really like the sound of, and failed in adding them to my routine, that would mean I would struggle when I tried to quit [smoking]. Like I didn't have the confidence to do it. But if I chose the ones I liked, I would be more likely to give it a proper go, you know? Then if I did that and changed those routines and noticed a difference. I'd be more likely to quit, or at least think I could quit or even believe I could quit" (PID 015, Low to moderately dependent smoker, PSQI 8).*

Learnt beliefs developed from years of experience navigating quit attempts motivated suggestions of adaptions in the existing CBT-I structure to tailor components to a smoker's needs during such a challenging time. Participants described rumination when trying to get to sleep being further exacerbated during a quit attempt because of their current negative mood state. For many this was a multifaceted issue, as they were unable to stop undesirable smoking related and craving thoughts from entering this cycle. Consequently, almost all participants suggested combining smoking based cognitive restructuring and sleep based cognitive restructuring. Participants believed that the addition of smoking based cognitive restructuring in conjunction with sleep, would lead to a more effective intervention and make it more feasible during a quit attempt.

*"If I was going to do this alongside a quit attempt, I would want some support for that as well. Like, I really like the idea of restructuring your thoughts or whatever, but I think why wouldn't you do it for smoking and sleep. Not just sleep, I think the thoughts for sleep could really help me but if you added smoking thoughts into that as well, I think that would probably be much better. Killing two birds with one stone kind of thing" (PID 014, Low to moderately dependent smoker, PSQI 4).*

*"So, I feel like if you do one without the other, I don't think it would be effective, and I feel like if you are able to tackle both at the same time and in relation to each other [. . .] it would really help to redevelop the ideas that you have about particular habits and behaviours you have" (PID 013, Low to moderately dependent smoker, PSQI 11).*

## Personalisation and digital delivery helping overcome psychological barriers during cessation

This theme highlights the role personalisation and digital delivery could play in reducing barriers to engagement. Digital delivery was viewed as being fluid, potentially allowing participants to build an intervention that co-exists with their personal journey of a quit attempt to facilitate autonomy. Recognition of the fragility of a quit attempt highlighted the value of an intervention design that allowed them to go at their own pace. Ease of access and convenience of logging onto a smartphone application or website, saving the time and money of travelling and fitting the intervention around their individual schedule were all barriers believed to be overcome by digital delivery. Perhaps the most important perceived benefit of remote delivery at a time of psychological vulnerability was the potential reduction of social anxiety or intimidation associated with attending face to face appointments. Not only did this appear applicable to interventions in general, but the reduction in stress may aid a quit attempt. Anything that attenuated anxiety in a time that was perceived as high stress, was seen as a potential factor in avoiding relapse.

*"For me personally I would find that way less awkward and I think a lot of other people would as well. Not that it's awkward but easier. And it's just convenience. To be honest I would definitely engage more if it was something I could get on my phone. I find going to things face to*

*face intimidating sometimes and often inconvenient" (PID 014, Low to moderately dependent smoker, PSQI 4).*

*"But I think if you're the other end of a screen, whether that's with a person or it's a fully auto-mated thingy, I think absolutely that would work and you don't have to trek into town, go for an appointment, work yourself up into an anxiety about that. It's just there in front of you. [. . .] Face to face can be a bit intimidating" (PID 017, Moderately dependent smoker, PSQI 5).*

Smoking cessation was seen as personal endeavour that cannot be generalised by a one size fits all approach. This was reflected by a lack of enthusiasm for the prescriptive approach of CBT-I. Empowerment and autonomy were identified as key for individuals using CBT-I as an aid to cessation. A systematic approach to personalisation and tailoring of the intervention by allowing more choice was a suggested method of achieving this. This was important both in terms of the intervention not being seen as generic and tailoring the intervention to the individ-ual's needs, allowing them to select which CBT-I components they want to engage in. If individ-uals were encouraged to engage in components, they did not believe in or were perceived as irrelevant to themselves, it would likely lead to a lack of engagement in all components.

*"You know I don't think there is a general. I don't think you can generalise this, I think it's quite situational and personal" (PID 009, Low dependent smoker, PSQI 4).*

*"So maybe if you could kind of choose the ones that, you either feel are most important or you feel like you would be able to do the best and then you know gradually increase the things that you do. But I think definitely if you could choose what they were" (PID 005, Low to moder-ately dependent smoker, PSQI 8).*

While the one size fits all approach was not seen as feasible, the overall the evaluation of CBT-Is potential effectiveness during a quit attempt was positive. Trust, confidence, and belief in the intervention was key in shaping views regarding acceptability and feasibility. Partici-pants generally wanted to try CBT-I components to aid a quit attempt primarily because they saw the benefits of not only giving up smoking but getting into a positive routine. A feeling of desperation among many, drove their desire to use components of CBT-I as they saw them as something that could really help.

*"I would give them all a go. As much as I could. I couldn't say, I'm going to stick to it 100% due to other factors such as I just mentioned but I want to stop smoking so I'm prepared to try anything to be honest" (PID 007, Moderately dependent smoker, PSQI 5).*

*"There were lots and lots of things on there that I would be so willing to try and do. I want to use this [. . .] this is a challenge I would love to take on and do. Because I can see the benefits of it (a) for me giving up smoking and (b) in my sleep getting a sleep routine [. . .] yeah okay there were a couple of little things which I didn't think were relevant to me, but actually a majority of that are changes I can make into my life" (PID 008, Low to moderately dependent smoker, PSQI 5).*

## Discussion

We identified four themes which reflect previous experiences and views of CBT-I to aid cessa-tion from the perspective of intervention target users. During quit attempts smokers experi-enced a range of sleep problems. The most commonly reported problems were frequent awakenings at night, difficulty returning back to sleep after awakenings, an increase in sleep

onset latency (SOL) and shorter sleep duration. Participants reflected that poor sleep during a quit attempt either had or could increase risk of relapse during cessation due to a negative impact on psychological state.

Participants thought that CBT-I is something they would use during a quit attempt to aid cessation but suggested changes and additions that would make engagement more feasible and tailored to quitting smokers. Key changes included the inclusion of smoking based cognitive restructuring, starting the intervention prior to a quit attempt, and the need for personalisation and tailoring. Overall, participants thought CBT-I was feasible and would be helpful during a quit attempt. The majority of participants wanted to use components of the intervention during their next quit attempt. However, participants also identified some potential unintended consequences such as failing to adhere to CBT-I could potentially impact self-efficacy and therefore engagement in a quit attempt.

Our findings build on existing literature investigating the association between sleep and smoking, including smoking cessation. Sleep problems described by participants in this study have been reported in previous observational and experimental research, such as a reduction in sleep duration [31], more frequent awakenings [32,33], increased SOL [34] and greater daytime fatigue [35]. In line with observational evidence [32], sleep issues reported in our study varied in intensity and duration between participants and persisted for up to three months post cessation. Poor sleep quality prior to [36–38] and during cessation [39] has also been shown to reduce the likelihood of a successful quit attempt. Our participants also suggested that poor sleep had, or could increase relapse risk.

Our study probed the perceived underlying mechanisms from a smoker's/ex-smokers' perspective. We found that smokers felt they were more vulnerable to making bad decisions or being more impulsive while having poor sleep during a quit attempt. This adds context to previous work that individuals with poor sleep quality, discount rewards more heavily in delay discounting tasks than those with better sleep quality [8], which has been found to be a strong prognostic indicator of smoking relapse [40]. Psychological states such as daytime fatigue, poor mood and poor cognition because of poor sleep also contributed to perceived risk of relapse in our study. All of which have been hypothesised as underlying mechanisms of the relationship between poor sleep and relapse vulnerability [5,9]. Our work combined with existing evidence suggest cognition, affect and emotion are plausible pathways in which poor sleep can impact smoking behaviour.

This body of evidence further validates sleep as a potential modifiable target to aid cessation. Few interventions have targeted sleep in quitting smokers. Those that have used CBT-I, or elements of, to improve sleep outcomes have showed evidence of promise [10,11] and good acceptability [10]. To the authors knowledge this is the first qualitative study to explore smokers and ex-smokers' perceptions of CBT-I as a smoking cessation aid. Our findings suggest that smokers and ex-smokers believe CBT-I would be more effective in aiding cessation if it started before a quit attempt. There is evidence that new habit formation is predictive of lower lapse risk [41], our work supports this, as participants felt adapting to CBT-I recommendations would not only boost their self-efficacy prior to cessation but also aid behaviour change. However, to achieve this, CBT-I should be flexible both in content and delivery because failure to adhere to CBT-I recommendations may mean individuals are less confident in their ability to successfully quit smoking. Which may reduce the likelihood of engaging in a quit attempt. Being able to select components of CBT-I based on personal preference and perceived value, provides a greater degree of autonomy, and increases the likelihood of perceived competence if an individual is successful in changing their behaviour around sleep prior to a quit attempt. Self-determination theory suggests that both a sense of autonomy and competence are essential for self-regulation and a sustained change in behaviour [42] and if these can be achieved

prior to a quit attempt, this may improve likelihood of cessation or quit attempt engagement. Further research is needed to explore the balance of autonomy and engagement with "active ingredients" of CBT-I as a bias towards one may have unintended consequences. One such solution to this problem is tailoring support based on clinical evidence but provide patients autonomy within that framework.

To promote successful behaviour change, it is essential that barriers to engagement are understood. Frequently cited barriers to intervention engagement in smokers are stigma [43], work and time constraints, access to resources, cost, travel [44] and low confidence [45]. Our findings suggest that digital delivery of CBT-I may overcome many of these barriers for smokers and reduce factors that may further decline mental health during cessation by removing the financial burden and stress associated with face-to-face appointments. Fully automated digital delivery of CBT-I has been shown to be effective within the general population [19] and we identified that this is neither a barrier to engagement for our participants and is unlikely to affect perceived effectiveness.

Participants identified components of CBT-I that may be unrealistic during a quit attempt. An effective sleep intervention to aid cessation would need to address these. First, sleep restriction was identified as being problematic for quitting smokers. Sleep restriction is often claimed to be the most effective standalone component of CBT-I [46]. It involves systematically reducing time spent in bed with the aim of initially matching the time in bed with reported sleep time in order to reduce night time awakenings, SOL and consolidate sleep [47]. In theory the more time a quitting smoker is out of bed, the more cravings they may experience and the more time they have to try to actively avoid smoking. While potentially problematic for quitting smokers, complete removal of this component may dilute the intervention and that needs to be carefully considered in any tailored CBT-I package.

Second, we found that the sleep hygiene/stimulus control recommendation of removing screen-based devices, in particular televisions, from the bedroom or night-time routine may have unintended consequences for participants coping strategies such as distraction during cessation. Despite previous work suggesting that disengagement coping strategies such as distraction may be less effective than others [41], various digital interventions targeting smoking cessation include or are built around distractions from craving[48,49] and both the utility and user preference for distraction components in such apps have been reported in previous qualitative work [50]. Although, there is a wealth of evidence of a negative association between screen time and sleep quality, CBT-I interventions, in the context of a quit attempt should consider the role distraction may play in individuals coping strategies.

Many CBT-I interventions that have tailored content to specific populations have included changes or additions based on recommendations from target users [51], which have subsequently improved engagement. We have found that the addition of smoking based cognitive restructuring to the existing sleep maladaptive thought restructuring in traditional CBT-I, increased perceived effectiveness and perceived likelihood of intervention engagement within the context of a smoking quit attempt. Cognitive restructuring for smoking cessation has previously employed problem solving, and coping skills embedded in relapse prevention theory. However, similarly to CBT-I the core focus is on changing maladaptive thoughts and existing literature suggests it is an effective smoking cessation treatment [52]. Although less research has focussed on the scalability of CBT for smoking cessation compared to CBT-I [52], based on the results of our study it is a necessary addition to CBT-I interventions hoping to aid cessation and should be considered in future work.

## Strengths and limitations

This study has several strengths, including the recruitment of a diverse range of smokers ranging from low to high levels of nicotine dependence and a nearly equal representation of males and females. However, there was limited ethnic diversity within the sample. A notable limitation is the reliance on describing CBT-I via a pre-prepared video. Participants feedback on intervention components may have been different if participants had experienced the intervention.

The majority of our participants reported poor sleep quality within the previous month which may make them more receptive to CBT-I as an adjunct to smoking cessation than a smoker with good sleep quality. We did not ask questions regarding other sleep related issues (e.g., Isolated Insomnia), which may have impacted sleep during an individual's quit attempt. Furthermore, while mental health during a quit attempt was referenced by participants, we did not collect information regarding current and/or past mental health conditions. Given the relationship between mental health, sleep and smoking, this is an important line of enquiry for future research and may have implications on tailoring CBT-I to quitting smokers. In addition, our sample only contained two participants with high nicotine dependence, limiting the transferability of findings to highly dependent smokers. Furthermore, Interviews took place online, suggesting that participants that took part in the study may have a higher digital literacy than others within our target demographic. Smokers with a lower digital literacy may reflect differently on digital delivery of CBT-I.

## Conclusion

This study provides valuable insight on smoker's and ex-smoker's experiences of sleep during cessation and perceived acceptability and feasibility of CBT-I as an aid to cessation. Findings highlight the need to tailor CBT-I to smokers if it is going to be utilised as an adjunct to aid cessation, and recommendations on how to do this are offered. CBT-I was perceived as both acceptable and feasible by smokers, dependent on tailoring to meet needs such as including smoking based cognitive restructuring and implementing CBT-I prior to a quit attempt.

## Supporting information

**S1 File. Additional information.**
(PDF)

**S1 Table. Full participant demographic table.**
(PDF)

## Acknowledgments

We are grateful to the people that participated in this study.

## Author Contributions

**Conceptualization:** Joe A. Matthews, Angela S. Attwood.

**Data curation:** Joe A. Matthews.

**Formal analysis:** Joe A. Matthews, Victoria R. Carlisle, Emma J. Dennie.

**Funding acquisition:** Joe A. Matthews, Angela S. Attwood.

**Investigation:** Joe A. Matthews.

**Methodology:** Joe A. Matthews, Victoria R. Carlisle, Robert Walker, Angela S. Attwood.

**Project administration:** Joe A. Matthews.

**Resources:** Joe A. Matthews.

**Supervision:** Ryan McConville, Hanna K. Isotalus, Angela S. Attwood.

**Writing – original draft:** Joe A. Matthews.

**Writing – review & editing:** Joe A. Matthews, Victoria R. Carlisle, Robert Walker, Claire Durant, Hanna K. Isotalus, Angela S. Attwood.

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
