## [Decision Letter · Decision Letter 0]

13 Dec 2023

PONE-D-23-21923“The worst thing is lying in bed thinking ‘I want a cigarette’” Exploration of smoker’s and ex-smoker’s perceptions of sleep during a quit attempt and the use of cognitive behavioural therapy for insomnia to aid cessation.PLOS ONE

Dear Dr. Matthews,

Thank you for submitting your manuscript to PLOS ONE. After careful consideration, we feel that it has merit but does not fully meet PLOS ONE’s publication criteria as it currently stands. Therefore, we invite you to submit a revised version of the manuscript that addresses the points raised during the review process.

Please pay close attention to comments left by Reviewer 2 and 3, who recommended Major revisions. ==============================

We look forward to receiving your revised manuscript.

Kind regards,

Lakshit Jain, MD

Academic Editor

PLOS ONE

Journal Requirements:

Reviewers' comments:

Reviewer's Responses to Questions

**Comments to the Author**

1. Is the manuscript technically sound, and do the data support the conclusions?

Reviewer #1: Yes

Reviewer #2: Yes

Reviewer #3: Yes

Reviewer #4: Yes

Reviewer #5: Yes

2. Has the statistical analysis been performed appropriately and rigorously? 

Reviewer #1: Yes

Reviewer #2: I Don't Know

Reviewer #3: N/A

Reviewer #4: Yes

Reviewer #5: I Don't Know

3. Have the authors made all data underlying the findings in their manuscript fully available?

Reviewer #1: Yes

Reviewer #2: No

Reviewer #3: Yes

Reviewer #4: Yes

Reviewer #5: Yes

4. Is the manuscript presented in an intelligible fashion and written in standard English?

Reviewer #1: Yes

Reviewer #2: Yes

Reviewer #3: Yes

Reviewer #4: Yes

Reviewer #5: Yes

5. Review Comments to the Author

Reviewer #1: The paper exhibits a well-defined and conventional academic structure, adhering to the customary sequence of introduction, methodology, and findings. This organizational approach enhances reader comprehension. Notably, the authors skillfully pinpoint a gap in existing research, offering a coherent rationale for their study. This strategic identification of a research gap contributes to establishing a clear purpose and direction for the research endeavor. Moreover, the paper excels in linking the research questions directly to the themes derived from data analysis. This connection underscores the relevance of the identified themes in addressing the overarching objectives of the study.

Few suggestions for improvement:

The paper should clearly articulate the implications of the findings to existing literature and real-world application. This includes addressing the potential consequences of ailing to adhere to interventions like CBT-I during smoking cessation attempts.

The introduction of the COVID-19 pandemic in “strengths and limitation" at the end and its impact on the participants' smoking and sleep patterns seems disjointed, suggesting a need to either provide sufficient contextual relevance or integration within the main discussion.

Overall, the paper endeavors to fill a gap in the literature by concurrently assessing the psychological impact of quitting smoking on sleep and investigating how behavioral therapy could alleviate those sleep disturbances, potentially improving cessation outcomes.

Reviewer #2: •Only one in 61 twenty smokers who engage in a quit attempt will remain smoke free [3] and 80% of quit 62 attempts will end in relapse within six months.

•Methods: age, sex, current smoking status, ethnicity, highest qualification attained, smoking history, quit-attempt history, e-cigarette use, cessation medication during quit attempts and cigarettes smoked per day. (In addition to this I would have included weight, history of snoring and past/ongoing psych history)

•Result: were mostly female (n= 9) –This was a bit strange (Study feels skewed and rationale for only 20 participants not mentioned)

•Potentially suggesting a prioritization of short-term mental wellbeing over longer-term health impacts. (Highlights the physical dependce of smoking: which the most problematic one)

•Timing of CBT-I and doing it a month before (This was a good suggestion, graded way to quitting smoking, it would be helpful for someone who has less motivation/less will power.

•I find going to things face to face intimidating sometimes and often inconvenient (Highlights digitalization- good for type c and type a people, who might smoke to reduce their anxiety)

•So participants has problem in all phases of sleep (early, middle, and late as well as initiating and maintaining)

•Discussion: Including the recruitment of a diverse range of smokers ranging from low to high levels of nicotine dependence and a nearly equal representation of males and females. (Its biased and only two participants with HIGH nicotine dependence)

•Well highlighted limitations, other than covid causing sleep disturbances, other bio-psycho-social factors should be considered. Isolated insomnia is very common.

•Overall, I feel the theme and aim of the paper is great. Validating their questionnaire, more participants, quantitive results using linear regression, incorporating med and psych factors can make this study robust. Lot of potential to improve.

Reviewer #3: I have read with great interest the article titled '“The worst thing is lying in bed thinking ‘I want a cigarette’” Exploration of smoker’s and ex-smoker’s perceptions of sleep during a quit attempt and the use of cognitive behavioural therapy for insomnia to aid cessation'. The article is well structured and results are presented in a clear concise manner. It presented insights into a very common problem noted when pt's attempt to quit smoking. CBT-I can be a good intervention to use for sleep disturbances associated with quitting smoking. While the study has many strengths there are limitation many of which have been enumerated by the authors. Sleep disturbances can be because of multiple causes. Most commonly sleep disturbances can be because of psychiatric disorders like depression, anxiety, other substance uses etc, medical disorders like thyroid disorders, pain, OSA etc. It is not clear whether these were considered in the inclusion/exclusion criteria. If they were considered, please mention what methods were used to evaluate/assess these conditions. If they were not considered, then it's a major limitation of the study that needs to be appropriately addressed. Please revise accordingly

Reviewer #4: Thank you for the opportunity to review this manuscript that describes a qualitative study looking at the perceptions of sleep among smokers and ex smokers and explores the utility of CBT-I to aid sleep and cessation.

-The background section was overall well written; could be condensed a little bit more.

-In terms of recruitment for the study, could the authors provide more details around whether the participants had a previous understanding/experience of CBT-I or CBT in general. Was this asked for ? The authors do highlight the reliance on using a pre-prepared video as a drawback.

-I think the authors mean ‘vicious’ and not viscous ?

-The results add to the literature by offering ideas on tailoring CBT-I for this purpose.

Overall good study and makes a good addition to the literature.

Reviewer #5: The manuscript by Matthews et al. studies the important topic of smoking cessation, predictors and strategies to prevent relapse.

Strengths of the paper:

- The title is appropriate for the content of the manuscript.

- The introduction to the article is well laid out.

- The methods section clearly describes the study design, subject selection, sample size, etc.

- The discussion and conclusion further elaborate on the study objectives and limitations.

6. PLOS authors have the option to publish the peer review history of their article (what does this mean?). If published, this will include your full peer review and any attached files.

Reviewer #1: **Yes: **Aditi Sharma

Reviewer #2: No

Reviewer #3: **Yes: **Nikhil Tondehal

Reviewer #4: No

Reviewer #5: No

---

## [Author Response · Author response to Decision Letter 0]

18 Jan 2024

Dear Lakshit Jain, MD,

Re: Manuscript ID PONE-D-23-21923

Thank you for the opportunity to revise and resubmit our article to PLOS ONE. 

We have addressed the reviewers’ comments, and we believe that these revisions have improved the paper by clarifying points and adding further detail about potential limitations. 

Please find below our responses. Changes are tracked in the revised manuscript as requested, alongside a ‘clean’ version of the manuscript. 

Thank you again for considering this revised version.

Yours sincerely,

Joe Matthews

School of Psychological Science

Response to editors’ requests

Thank you for this information, I can confirm that the manuscript adheres to PLOS ONE’s style requirements.

This submission does not have any author generated code as this is not a study using quantitative methods. However, author generated themes from the qualitative analysis are detailed within the manuscript.

Thank you for this suggestion. Our School champions Open Science practices. This study was pre-registered on the open science framework and upon acceptance, anonymised interview transcripts will be uploaded to Bristol University’s file repository for other researchers to access.

Thank you for this information. This aligns with our open research protocols and agree should this manuscript be accepted, it should be held until the relevant DOI is provided.

Response to Reviewers’ comments

Reviewer 1

1. The paper exhibits a well-defined and conventional academic structure, adhering to the customary sequence of introduction, methodology, and findings. This organizational approach enhances reader comprehension. Notably, the authors skillfully pinpoint a gap in existing research, offering a coherent rationale for their study. This strategic identification of a research gap contributes to establishing a clear purpose and direction for the research endeavor. Moreover, the paper excels in linking the research questions directly to the themes derived from data analysis. This connection underscores the relevance of the identified themes in addressing the overarching objectives of the study. 

We thank reviewer 1 for their positive comments on the study and the manuscript.

2. The paper should clearly articulate the implications of the findings to existing literature and real-world application. This includes addressing the potential consequences of ailing to adhere to interventions like CBT-I during smoking cessation attempts.

We agree this is important and is something we are planning to research within the future. There is currently no existing evidence about the potential consequences of not adhering to CBT-I on a quit attempt outcome. However, we have now included the following sentence discussing potential consequences of not adhering to the quit attempt on self-efficacy and future quit attempt engagement on line 515, pg.25 “However, to achieve this, CBT-I should be flexible both in content and delivery because failure to adhere to CBT-I recommendations may mean individuals are less confident in their ability to successfully quit smoking. Which may reduce the likelihood of engaging in a quit attempt”.

3. The introduction of the COVID-19 pandemic in “strengths and limitation" at the end and its impact on the participants' smoking and sleep patterns seems disjointed, suggesting a need to either provide sufficient contextual relevance or integration within the main discussion.

After review, we have decided to remove this from the “strengths and limitations” section. 

Reviewer 2

1. Methods: age, sex, current smoking status, ethnicity, highest qualification attained, smoking history, quit-attempt history, e-cigarette use, cessation medication during quit attempts and cigarettes smoked per day. (In addition to this I would have included weight, history of snoring and past/ongoing psych history)

Thank you for these suggestions. We will consider including these measures within future research. 

2. Result: were mostly female (n= 9) –This was a bit strange (Study feels skewed and rationale for only 20 participants not mentioned)

This was a qualitative study that recruited openly and without minimisation or balancing for biological sex. While, there were more females (n=9), males were represented (n=8) and we do not feel the difference is substantial enough to undermine the findings. We did not have strong theoretical expectations of differences between sexes, so feel this is adequate for the question and reasonable balanced for this methodology.

We agree, our recruitment target should be justified. This was based on data saturation expectations, and we now provide a reference to support this approach (148 pg. 6). This paper provides recommendations for recruitment and samples sizes in qualitative research. 

3. Discussion: Including the recruitment of a diverse range of smokers ranging from low to high levels of nicotine dependence and a nearly equal representation of males and females. (Its biased and only two participants with HIGH nicotine dependence)

We have included this statement within the strengths and limitations section “In addition, our sample only contained two participants with high nicotine dependence, limiting the transferability of findings to highly dependent smokers” Line 590 pg. 28.

4. Well highlighted limitations, other than covid causing sleep disturbances, other bio-psycho-social factors should be considered. Isolated insomnia is very common.

We thank the reviewer for highlighting this. Isolated insomnia has now been described and detailed as a potential limitation on line 589 pg. 28 in the following statement “We did not ask questions regarding other sleep related issues (e.g., Isolated Insomnia), which may have impacted sleep during an individual’s quit attempt”.

5. Overall, I feel the theme and aim of the paper is great. Validating their questionnaire, more participants, quantitive results using linear regression, incorporating med and psych factors can make this study robust. Lot of potential to improve.

We think the reviewer may have misunderstood the research design. This is a qualitative study that used semi-structured interviews to collect prose (i.e., final data set was a text-based transcript). We did not produce or deliver a text-based questionnaire or validate a questionnaire, and therefore there is no quantitative data upon which to run a linear regression. To make the study design clearer, we have amended the title to include “qualitative” see line 2 pg.1 (““The worst thing is lying in bed thinking ‘I want a cigarette’” A qualitative exploration of smoker’s and ex-smoker’s perceptions of sleep during a quit attempt and the use of cognitive behavioural therapy for insomnia to aid cessation).

While we agree, quantitative methods are important, there is a wealth of evidence that has determined a relationship between sleep, smoking and smoking cessation using such methods (Patterson et al. 2019 https://academic.oup.com/ntr/article/21/2/139/4562639). The primary aim of this study was to understand from a smoker’s perspective if a sleep intervention could be used to mitigate such relationships.

Reviewer 3

1. I have read with great interest the article titled '“The worst thing is lying in bed thinking ‘I want a cigarette’” Exploration of smoker’s and ex-smoker’s perceptions of sleep during a quit attempt and the use of cognitive behavioural therapy for insomnia to aid cessation'. The article is well structured and results are presented in a clear concise manner. It presented insights into a very common problem noted when pt's attempt to quit smoking. CBT-I can be a good intervention to use for sleep disturbances associated with quitting smoking. 

We thank the reviewer for their kind comments and agree there is potential to use CBT-I to aid sleep during cessation.

2. While the study has many strengths there are limitation many of which have been enumerated by the authors. Sleep disturbances can be because of multiple causes. Most commonly sleep disturbances can be because of psychiatric disorders like depression, anxiety, other substance uses etc, medical disorders like thyroid disorders, pain, OSA etc. It is not clear whether these were considered in the inclusion/exclusion criteria. If they were considered, please mention what methods were used to evaluate/assess these conditions. If they were not considered, then it's a major limitation of the study that needs to be appropriately addressed. Please revise accordingly.

We agree with the reviewer that psychiatric disorders are very interesting fields of enquiry but understanding the impact of these on sleep during a quit attempt was not one of the aims of this research. The primary aim of this research was to understand smoker’s perspectives of the potential use CBT-I during a quit attempt to improve sleep. The secondary aim was to understand smoker’s perspectives of sleep during a quit attempt. Interviews used open ended questions and participants were asked what impacts their sleep generally, at no point did any participants suggest other medical or psychiatric conditions impacted their sleep or would prevent them from engaging with CBT-I. As this study was aiming to understand the perspectives of smokers, not infer causation or associations (as there is already evidence to this effect, see Patterson et al. 2019 https://academic.oup.com/ntr/article/21/2/139/4562639) we did not feel it was necessary to exclude individuals with other disorders or conditions. However, given the higher prevalence of both smoking and sleep problems within mental health conditions, this is an important line of enquiry that will need to be considered and tested within any future intervention development. We have added acknowledgement of this in the discussion on line 591 pg.28 “Furthermore, while mental health during a quit attempt was referenced by participants, we did not collect information regarding current and/or past mental health conditions. Given the relationship between mental health, sleep and smoking, this is an important line of enquiry for future research and may have implications on tailoring CBT-I to quitting smokers”.

Reviewer 4

1. In terms of recruitment for the study, could the authors provide more details around whether the participants had a previous understanding/experience of CBT-I or CBT in general. Was this asked for ? The authors do highlight the reliance on using a pre-prepared video as a drawback.

Thank you for highlighting this. Understanding/ experience of CBT-I was not asked during recruitment and was not part of inclusion/ exclusion criteria. However, participants were asked this as part of the interview. We have now included the following statement on line 223 pg. 10: “None of the study participants were aware of or had experience of CBT-I prior to the interview”.

2. I think the authors mean ‘vicious’ and not viscous ?

Thank you for highlighting this error. This has now been corrected.

3. The results add to the literature by offering ideas on tailoring CBT-I for this purpose.

Overall good study and makes a good addition to the literature.

We thank the reviewer for their positive comments regarding this manuscripts contribution to the literature.

Reviewer 5

1. Strengths of the paper:

- The title is appropriate for the content of the manuscript.

- The introduction to the article is well laid out.

- The methods section clearly describes the study design, subject selection, sample size, etc.

- The discussion and conclusion further elaborate on the study objectives and limitations.

We thank reviewer 5 for their positive words regarding our study and manuscript.

---

## [Decision Letter · Decision Letter 1]

15 Feb 2024

“The worst thing is lying in bed thinking ‘I want a cigarette’” A qualitative exploration of smoker’s and ex-smoker’s perceptions of sleep during a quit attempt and the use of cognitive behavioural therapy for insomnia to aid cessation.

PONE-D-23-21923R1

Dear Dr. Matthews,

We’re pleased to inform you that your manuscript has been judged scientifically suitable for publication and will be formally accepted for publication once it meets all outstanding technical requirements.

Kind regards,

Lakshit Jain, MD

Academic Editor

PLOS ONE

Additional Editor Comments (optional):

Reviewers' comments:

Reviewer's Responses to Questions

**Comments to the Author**

1. If the authors have adequately addressed your comments raised in a previous round of review and you feel that this manuscript is now acceptable for publication, you may indicate that here to bypass the “Comments to the Author” section, enter your conflict of interest statement in the “Confidential to Editor” section, and submit your "Accept" recommendation.

Reviewer #1: All comments have been addressed

Reviewer #5: All comments have been addressed

2. Is the manuscript technically sound, and do the data support the conclusions?

Reviewer #1: Yes

Reviewer #5: Yes

3. Has the statistical analysis been performed appropriately and rigorously? 

Reviewer #1: Yes

Reviewer #5: I Don't Know

4. Have the authors made all data underlying the findings in their manuscript fully available?

Reviewer #1: Yes

Reviewer #5: Yes

5. Is the manuscript presented in an intelligible fashion and written in standard English?

Reviewer #1: Yes

Reviewer #5: Yes

6. Review Comments to the Author

Reviewer #1: I am pleased to acknowledge that the authors have effectively addressed all the comments provided by the reviewers in their revised manuscript. The meticulous attention to feedback is evident in the improved clarity, strengthened arguments, and enhanced overall quality of the content. The authors' responsiveness reflects their commitment to producing a high-quality piece of work. With these commendable revisions, I believe the manuscript is now well-prepared for the next stage of the review process.

Reviewer #5: The authors have adequately answered the questions raised by the reviewers, and it is now suitable for publication.

7. PLOS authors have the option to publish the peer review history of their article (what does this mean?). If published, this will include your full peer review and any attached files.

Reviewer #1: No

Reviewer #5: No

---

## [Editor Report · Acceptance letter]

26 Apr 2024

PONE-D-23-21923R1 

PLOS ONE

Dear Dr. Matthews, 

I'm pleased to inform you that your manuscript has been deemed suitable for publication in PLOS ONE. Congratulations! Your manuscript is now being handed over to our production team.

Kind regards, 

on behalf of

Dr. Lakshit Jain 

Academic Editor

PLOS ONE